# Changes in Eating Behaviors during the COVID-19 Lockdown and the Impact on the Potential Inflammatory Effects of Diet

**DOI:** 10.3390/ijerph19159079

**Published:** 2022-07-26

**Authors:** María del Pilar Montero López, Ana Isabel Mora Urda, Francisco Javier Martín Almena, Oscar Geovanny Enríquez-Martínez

**Affiliations:** 1Departamento de Biología, Facultad de Ciencias, Universidad Autónoma de Madrid, 28049 Madrid, Spain; 2Departamento de Didácticas Específicas, Facultad de Formación del Profesorado, Universidad Autónoma de Madrid, 28049 Madrid, Spain; ana.mora@uam.es; 3Facultad de Ciencias de la Salud, Universidad Católica de Ávila, 05005 Avila, Spain; fjavier.martin@ucavila.es; 4Public Health Program, Health Sciences Center, Federal University of Espirito Santo, Vitória 29075-910, Brazil; oscar.enriquez.m@outlook.com

**Keywords:** COVID-19, eating behavior, lockdown, anti-inflammatory diet, gender

## Abstract

Background: This cross-sectional study compares eating behaviors before and during the COVID-19 lockdown that was decreed in Spain on 14 March 2020. Methods: The sample was made up of 1177 people aged 18 years or older who responded during the month of June 2020 to a questionnaire designed in Google Forms. Information was collected on the frequency of food consumption before and during lockdown. A dietary inflammatory index (DII) was created with positive or negative values depending on the inflammatory potential of different foods, vegetables, fruits, nuts, legumes, meat, fish, eggs, yogurt, milk, cheese, industrial pastries, salty snacks, fast food, and soft drinks. The scores from before and during confinement were compared. Results: Most of the people in the sample maintained their eating pattern during lockdown. Among those who changed, the majority increased their consumption of healthy foods, which resulted in a decrease in the inflammatory potential of the diet; this was particularly the case in men. Conclusions: The improvement in the quality of the diet contributed to a significant decrease in DII during confinement, especially in men.

## 1. Introduction

At the end of 2019, cases of pneumonia of unknown origin were detected in Wuhan, China and spread very quickly. After some research, certain similarities were found with previous epidemics, such as severe acute respiratory syndrome (SARS-CoV) in 2003 and Middle East Respiratory Syndrome (MERS) in 2012. The virus identified in 2019 was called Severe Acute Respiratory Syndrome Coronavirus 2 (SARS-CoV-2), and the disease was called coronavirus disease 2019 (COVID-19) [1].

The World Health Organization (WHO) decreed COVID-19 as a pandemic on 11 March 2020 due to the rapid and progressive worldwide spread of the virus and asked political leaders of countries to adopt prevention measures [2]. Since the declaration of a global health emergency by the WHO, each government has managed the situation using different strategies. The measures adopted included mandating total confinement, maintaining normality to obtain so-called “collective immunity”, going through selective quarantines for part of the population, and encouraging total confinement for vulnerable people (the elderly and/or people with underlying diseases).

In Spain, on 13 March 2020, the government declared a state of emergency, which was enforced on 14 March and led to home confinement in the entire country to stop the transmission of COVID-19. The Royal Decree limited the freedom of movement of people during the period of the state of emergency. Citizens could use public thoroughfare to purchase necessities, attend a health or service center, work if essential activities could not be carried out electronically, and return to their habitual residence [3]. These measures were maintained for the first wave of the pandemic from 14 March to 21 June 2020.

The practices of total or partial confinement and physical social distancing have been and continue to be very important to prevent the collapse of the health system, but they undoubtedly affect many aspects of daily life, including the eating habits and lifestyles of individuals, families, and populations. These changes in the habitual behavior of the population, both in food consumption and in physical activity, can affect people’s health.

Dietary patterns are specific to sociodemographic variables and have been shown that food consumption varies in quantity and quality between men and women. International organizations and scientific journals insist on the importance of presenting the results disaggregated by gender [4,5,6].

In addition to the quantity, it is important to consider the quality of the food. In this sense, healthy eating patterns are associated with lower circulating concentrations of inflammatory markers [7].

A proinflammatory diet is one that favors increased levels of proinflammatory biomarkers, such as C-Reactive protein (CRP), interleukins 6 and 8 (IL-6 and IL-8, respectively), and tumor necrosis factor alpha (TNF-α) [8,9]. CRP is a key inflammatory marker that is produced in the local epithelium and the liver in response to certain cytokines and induces inflammatory processes by recruiting monocytes or mediating the absorption of low-density lipoproteins by endothelial macrophages. A proinflammatory diet is characterized by high levels of saturated fatty acids and simple sugars, which are present in certain food groups, such as meats, processed foods, and sugary drinks [9] and can have negative consequences on health, such as increasing the risk of obesity, hypertension, diabetes, and hypercholesterolemia [7,10]

The opposite effect is produced by foods classified as “anti-inflammatory” such as fish, fruits, vegetables, and legumes, which are rich in unsaturated fatty acids, fiber, vitamin C, and micronutrients such as calcium, phosphorus, and magnesium. An anti-inflammatory diet is associated with high levels of the anti-inflammatory markers interleukin 10 (IL-10) and tumor necrosis factor beta (TNF-β) [7,10,11]. A dietary inflammatory index (DII) was developed by Cavicchia et al. (2011) [12] to quantify the proinflammatory or anti-inflammatory potential of the diet. High positive scores indicate a mostly proinflammatory diet, and negative scores indicate a mostly anti-inflammatory diet.

This work will present some of the results obtained from the ALVIMED project regarding the impact of confinement on eating behaviors and diet quality in men and women. Our working hypothesis was that confinement would lead people to consume less healthy foods, thereby reducing the quality of their diet and increasing its inflammatory potential.

The objectives of this study were as follows:

To determine the impact of the COVID-19-related total and partial confinement between March and June 2020 on diet in a Spanish sample with a medium-high level of studies. The specific aims were to describe the changes in eating behavior during confinement, and to assess the effect of this change on the proinflammatory or anti-inflammatory properties of the diet. All these objectives were intended to be achieved separately according to gender.

## 2. Materials and Methods

It is a cross-sectional study, developed from 1 April to 30 June 2020, time of social distancing and restrictions caused by the first wave of COVID-19. The participants were adults over 18 years, residents in Madrid, Spain. An online questionnaire with closed questions was created, evaluating sociodemographic variables, eating habits before and during confinement, habits and lifestyle, anthropometric indicators, COVID Health.

In June 2020, the questionnaire was sent by the members of the research team to students, faculty, and the administration and services staff of the University Autónoma de Madrid, who in turn distributed it to at least 10 people in their close social circle. All participants participated voluntarily; they agreed to participate before starting to complete the survey.

The protocol was approved by the Research Ethics Committee of the Autonomous University of Madrid (Ref: CEI-106-2082).

### 2.1. Sample

A non-probabilistic sampling was used in this study, the link to the questionnaire was released for personal and institutional e-mail. Those who agreed to participate in the study completed an online structured questionnaire using the Google Forms web survey platform. A total of 1177 people 18 years of age or older responded to the questionnaire: 854 women (72.56%), 320 men (27.19%), and three individuals who did not specify their gender (0.25%). People under 18 years old and without residence in Spain were excluded.

### 2.2. Variables

Information on the following variables was collected in the survey:

Sociodemographics: Sex, age, educational level (primary, secondary, middle, and higher studies), occupation (skilled workers, low-skilled workers, students, housewives/unemployed, retired).

Eating behaviors before and during confinement: Information was collected on the number of daily meals (breakfast, mid-morning, noon, snack, dinner) and frequency of daily or weekly consumption of vegetables, fruit, legumes, nuts, meat, fish, eggs, milk, yogurt, cheese, industrial pastries, salty snacks, fast food, soft drinks, beer, and distilled alcohol before and during confinement. The questionnaire was based on the validated FFQ in a sample with similar characteristics [13]


*Frequency of Food Consumption before Lockdown and during Lockdown*


0: Never1: 1–2 times a week2: 3–4 times a week3: 5–6 times a week4: Once a day5: Two or more times a day


*Dietary Inflammatory Index (DII)*


A dietary inflammatory index was calculated following the indications of Cavicchia et al. (2009) [12] that determined the ‘inflammatory potential’ of different food and beverage components according to their predicted effects on C-reactive protein (CRP), a known biomarker of inflammation. Negative or positive values of 0, 1, or 2 were assigned to the consumption frequencies of each food. A negative sign indicates an anti-inflammatory effect, and a positive sign indicates a proinflammatory effect. To assign values of inflammation to the different categories of food consumption, those proposed by Davis et al. 2019 [10] were used:Anti-inflammatory foods: Vegetables, fruits, legumes, nuts, and fish. The values assigned to each frequency of consumption were as follows:

0 = never or sporadically

−1 = 3–6 times/week

−2 = 7 or more times/week.

Proinflammatory foods: Eggs, dairy products, industrial pastries, salty snacks, fast food, soft drinks. The values assigned to each frequency of consumption of these foods were as follows:

0 = never or sporadically

+1 = 3–6 times/week

+2 = 7 or more times/week.

A total DII score was calculated by adding the values of the inflammatory potential of each food according to the frequency of consumption. A total DII score was calculated for consumption before lockdown and another score was calculated for consumption during lockdown. In this way, both results could be compared to determine if they exhibited any variation.

Habits and lifestyle: In Spain, confinement was mandatory for 100% of the population. The questions for the practice of physical activity were self-reported and participants were asked whether during the period of confinement if they practiced some type of physical activity. The answer options were yes or no.

Anthropometric measures: Self-reported weight (kg) and self-reported height (m), with which the body mass index was calculated (BMI = weight (kg)/height (m^2^)).

COVID Health: It was also asked whether or not the person had suffered from COVID-19 and, if so, whether or not she or he were hospitalized for it.

### 2.3. Statistical Methods

With all these variables, a database was created that was statistically analyzed with the IBM Statistical Package for the Social Sciences 26.0. Before describing the variables and applying the statistical tests, the normal distribution of the quantitative variables was verified with the Kolmogorov–Smirnoff test. The only variables that did not fit the normal distribution were the kilograms gained and lost during lockdown.

The association between qualitative variables was analyzed with the chi-square test. The frequencies of consumption of the different types of food before and during lockdown were compared for women and men.

The concordance between frequencies of food consumption before and during lockdown (for the total sample and in women and men separately) was assessed using Cohen’s kappa test. This statistical test measures the agreement between responses for qualitative variables with the same number of categories.

According to the value of k, different levels of agreement strength are distinguished: poor if k is less than 0.20; weak if k is between 0.21 and 0.40; moderate if k is between 0.41 and 0.60; good if k is between 0.61 and 0.80; and very good if k is greater than 0.81 [14]. For this test, the null hypothesis is that there is no agreement, so *p* values lower than 0.05 would indicate that there is agreement between the responses obtained.

The comparison of the DII score between men and women was carried out with Student’s t-test for independent samples. Then, the comparison of the DII before and during lockdown was carried out using Student’s *t*-test for related samples in men and in women separately.

The analysis of the differences in DII between occupations was carried with the Kruskal–Wallis test. Finally, we analyzed the differences between DII scores before and during lockdown for each occupation with the Wilcoxon test.

In all analyses, the level of statistical significance was *p* < 0.05.

## 3. Results

The characteristics of the sample are shown in Table 1. The studied sample had a medium-high level of education. Most of the subjects in the sample had college or high school education. Almost 70% reported doing some type of physical activity at home during the lockdown. With regard to BMI, at the time they responded to the questionnaire, women had mean self-reported values corresponding to normal weight, and men had mean values situated in the lower limit of overweight.

The prevalence of those affected by COVID-19 was approximately 13%, but only approximately 1% had to be hospitalized.

### 3.1. Eating Behavior

Table 2 shows the results of the comparison of the frequency of consumption of vegetables, fruits, pulses, and dry fruits between men and women before and during lockdown.

The most common frequency of vegetable consumption both before and during lockdown was between 1 and 2 times and 3 or 4 times a week. Statistically significant differences were observed between women and men in the consumption of vegetables. In the categories of higher consumption (5 to 6 times/week and 1 or more times a day), there were more women than men both before and during lockdown. However, as seen in the table, men increased the frequency of consumption of vegetables during lockdown.

With regard to fruit, before confinement, most of the people consumed fruit 1 or 2 times/day. During lockdown, the majority remained in this category, but the percentage of people who consumed fruit more than 2 times per day increased.

Similar results were obtained for dry legumes; most people consumed them 1 or 2 times/week. However, during lockdown, the percentage of people who consumed them 3–4 times/week increased. Regarding the consumption of nuts, the highest percentage was found in the 1–2 times/week category, but during lockdown, the percentage of people who consumed nuts 3–4 times/week, 5–6 times/week, and even 1 time per day increased.

The same was found for the consumption of fish, except that during confinement, the percentage of people who never consumed it increased and the percentage of people who consumed it 5–6 times/week increased as well (Table 3).

During lockdown, the percentage of people who consumed eggs 5–6 times/week and once a day increased.

Statistically significant differences were observed between men and women only for the consumption of eggs. Before lockdown, there was a higher percentage of men than women who ate eggs once a day or more. For meat and fish, no statistically significant differences were observed between men and women (Table 3).

Again, regarding the concordance between the frequency of consumption before and during lockdown, the Kappa coefficient showed moderate-to-good agreement, which indicates that the consumption pattern was maintained.

Table 4 shows the results of the frequency of consumption of dairy products and the comparison between women and men before and during lockdown.

Regarding milk, the highest percentages of consumption were found for the 1–2 times/week and once a day, remaining similar before and during lockdown.

Most people consumed yogurt 1–2 times/week. During lockdown, the percentage of people who consumed it 3–4 times/week increased. The percentage of people who consumed it more than 2 times a day and who never consumed it also increased slightly.

Almost half of the people in the sample consumed cheese 1–2 times/week prior to the lockdown, but during lockdown, this percentage decreased somewhat. In addition, the percentage of people who never consumed cheese and of those who consumed it more than 5–6 times/week increased. The percentage of people who consumed cheese 3–4 times/week decreased. Between women and men, there were significant differences in the consumption of cheese, with a higher frequency of women who consumed cheese once a day or more, both before and during lockdown. During lockdown, the percentage of people who consumed once a day or more increased for both genders, but women continued to consume more. For milk and yogurt, no statistically significant differences were observed between men and women.

Regarding the concordance between the frequency of consumption before and during lockdown, the concordance was good, which indicates that the consumption pattern was maintained for all the foods in the table.

The frequencies of consumption of processed and industrial foods are shown in Table 5. More than 55% of the sample consumed fast food before lockdown, while during lockdown, the percentage of people who never consumed it increased. The same finding was obtained for industrial pastries. No statistically significant differences were observed between men and women for any of the foods included in this table, neither before nor during lockdown.

Regarding the agreement between the frequency of consumption before and during lockdown, the Kappa coefficient was rather low, which indicates that the consumption pattern was not maintained, especially for salty snacks, the consumption of which increased significantly, especially in women.

Table 6 shows the results of the consumption of soft drinks and alcohol. The most frequent consumption category for both fermented and distilled alcohol was once or twice a week before lockdown and never during lockdown. There were significant differences between men and women in the consumption of beer/wine and distilled alcohol. Before lockdown, men consumed both types of alcohol more frequently than women, and this pattern was maintained during lockdown. For soft drinks, a decrease was also seen, increasing the number of people who never drank them during confinement. No statistically significant differences were observed between men and women either before or during lockdown.

A moderate concordance was observed in general between consumption before and during lockdown, which indicates that the pattern was maintained. This concordance was found for both soft drinks and alcohol. Only in the case of distilled alcohol in women was the concordance rather low, which indicates that the consumption of distilled alcohol in women changed, and the percentage of women who never consumed distilled alcohol increased. The percentage of those who consumed it 5–6 times a week and once a day also increased very slightly.

### 3.2. Dietary Inflammatory Index (DII)

Table 7 shows the results of the DII values before and during lockdown.

The mean value of the DII both before and during lockdown in both sexes presented a negative value, which indicates an anti-inflammatory characteristic of the diet. Not significant differences are observed between women and men.

Before lockdown, the DII was more negative among women than among men; that is, in general, women followed a more anti-inflammatory diet than men. During lockdown, the DII became more negative for both genders, which reflects that the consumption of foods with anti-inflammatory potential increased, and the difference between the genders decreased.

Regarding the difference in the score before and during confinement, in women, no significant differences were observed for the total score between the two periods, but differences were observed for the values of some foods. During lockdown, the consumption of fruit, nuts, eggs, cheese, and salty snacks increased; however, that of soft drinks and fast food decreased.

During confinement, men increased their consumption of vegetables and decreased their consumption of soft drinks and fast food, which made the inflammatory potential of their diet decrease, and the mean score was lower with statistical significance, indicating that their diet changed to a more anti-inflammatory and healthy diet.

Overall, significant changes were observed in the consumption of vegetables, fruit, nuts, cheese, salty snacks, soft drinks, and fast food, with increases in the consumption of vegetables, fruit, nuts, and cheese during lockdown. The consumption of soft drinks and fast food decreased.

Other variables such as the age of the person and their occupation have also been able to influence eating behavior. For this reason, the median values of the DII before the lockdown and during the lockdown have been compared between the occupations of the participants (qualified professions, low-skilled workers, students, housewives/unemployed, and retired).

Figure 1 shows the median values by occupation obtained for the DII before and during confinement, and the difference between the two scores.

The comparison between the median values of DII for each occupation was analyzed with the Kruskal–Wallis test (Table 8). Statistically significant differences were observed between the four occupational groups considered, with housewives, unemployed people, and retired people showing the most anti-inflammatory food values both before and during confinement.

The DII decreases ostensibly during confinement for skilled workers, low-skilled workers, and students, that is, their diet became more anti-inflammatory. The most pronounced difference between the DII scores before and during lockdown occurs in students.

The results of the comparison between the DII scores before and during the confinement, carried out with the Wilcoxon test, show statistically significant differences for low-skilled workers (*p* = 0.020) and for students (*p* < 0.001), which indicate an improvement in the anti-inflammatory potential of their diet (Figure 1).

## 4. Discussion

There has been a great deal of interest in the effects that restricted mobility may have on the eating behavior and lifestyle of the population worldwide. For this reason, in the last year, numerous investigations have been carried out on this subject in different countries and population groups.

This study presents results on eating behavior during confinement due to COVID-19 in a non-probabilistic sample with a medium-high educational level. Therefore, it is a non-representative sample of the Spanish population in terms of educational level, which is also reflected in lower mean BMI values than those observed for the general population in both men and women [15]. The results regarding changes in eating behavior were similar to those found in other works conducted with samples similar to the one studied [15,16,17,18,19,20].

Our working hypothesis was that confinement would lead people to consume less healthy foods, thereby reducing the quality of their diet and increasing its inflammatory potential. At the beginning of the confinement, data provided by government ministries and institutions on the massive purchases of some beverages and food products indicated that the population’s diet was going to get worse [21]. However, later it was observed that in some sectors of the population this was not the case. The results obtained in this study coincide with other samples with a medium-high level of education, in which it has been proven that eating behavior improved [19,20,22]. In general, most people did not change their eating behavior, but a large part of those who did improved their diet, with an increase in the consumption of healthier foods such as vegetables and fruits, along with a decrease in the consumption of less-healthy foods such as industrial pastries and alcoholic beverages. In other words, during lockdown, the quality of the diet increased. These results coincide with those of studies carried out with samples of medium-high education level, in Latin American, Mediterranean, and other European countries [20,22,23].

The results obtained do not support these hypotheses since an improvement in eating behavior was observed, which leads to a decrease in DII. There are many possible reasons for this improvement; on the one hand, during confinement, families spent more time cooking, and as a consequence, there was a decrease in the consumption of processed foods. On the other hand, not being able to leave home could contribute to a decrease in consumption of fast food and alcohol, as was also observed in the results of other similar investigations [7,18,19,22,23].

The decrease in DII during confinement is related to the lower consumption of foods with proinflammatory potential, such as industrial and processed foods and alcohol, also observed in other Spanish samples. The DII decreased significantly, especially in men. Before confinement, the diet of men was of poorer quality than that of women, so the change was more evident, thus resulting in gender differences.

Women consumed more vegetables than men, which is in agreement with the results of Ruiz-Roso et al. (2020) [23]. However, no significant gender differences were observed for fruit consumption, while in the study by Ruíz-Roso et al. (2020) [23], gender differences were observed, with women consuming significantly more fruit during confinement. Another difference was found in higher soda consumption in men.

That is, gender differences were observed in the pattern of consumption of food and beverages; in general, women had healthier eating patterns, both before and during confinement [20].

These results also show that, in this sample, diet improved significantly in people who stopped going to their work and study centers and stayed at home due to lockdown. In fact, in the results by occupation, the greatest improvement, with significant differences between before and during confinement, is observed in workers and students who, unable to travel to their places of study and work, probably had more meals at home. In housewives, unemployed, and retired people, the change was less relevant.

This study has some limitations and strengths; it is a cross-sectional study, which leads to reverse causality.

The main limitation of our study is the lack of representativeness of the population sample. Moreover, all data collected are self-reported and this could make them not completely reliable. However, the special conditions in which the work was carried out must be considered—in Spain, there was a total lockdown for four months between March and June 2020. The results of our study support the idea, expressed by other authors, that this period of confinement could have been an opportunity to clearly demonstrate that it is possible to change eating habits in a population towards a healthier model with less consumption of fast food, soft drinks, and alcohol. However, we must not forget the disadvantaged population groups for whom the pandemic has meant a reduction in their work activity, including loss of their jobs, which has had a negative impact on access to certain foods. In a short time, COVID-19 has magnified already existing disparities in access to food in general and to healthy food in particular in low-income individuals and households [24]. The questionnaires were applied in a single moment and led the participants to remember their life habits before the pandemic, which may present memory bias. Diet information was not used to quantify food consumption, nutrients, or other components. Instead, such information is subject to the individual’s memory, which could have caused measurement errors inherent to this type of method [25].

## 5. Conclusions

The most relevant conclusions of this work would be:

No great change in eating behavior was observed, and most of the people in the sample maintained their eating pattern during lockdown.

In the people whose diets changed, the shift was mostly towards a healthier eating pattern, with a decrease in the consumption of fast food, soft drinks, and alcoholic beverages and an increase in the consumption of vegetables and fruits.

The decrease in the consumption of fast food, soft drinks, and alcoholic beverages may be related to a “social consumption” of these foods and beverages which was reduced by mobility restrictions and social gatherings.

The improvement in the quality of the diet contributed to a significant decrease in DII during confinement, especially in men.

Men had a more anti-inflammatory diet during confinement compared to pre-confinement.

## Figures and Tables

**Figure 1 ijerph-19-09079-f001:**
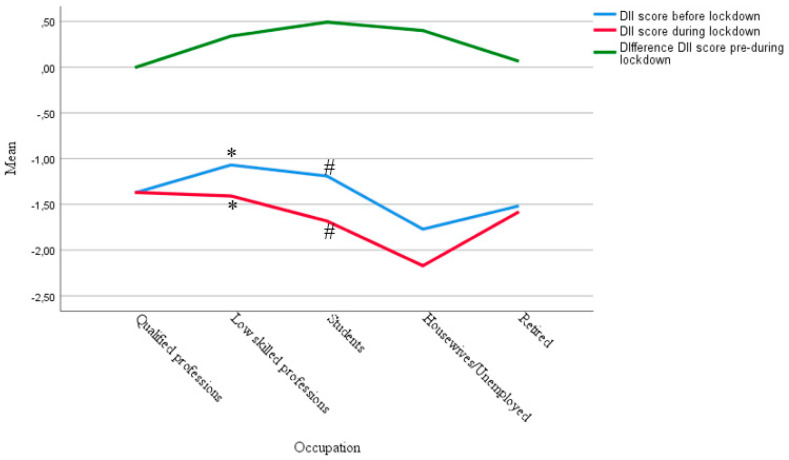
Differences on DII score between occupational groups; * Significant difference between during and before lockdown in low skilled professions; # Significant difference between during and before lockdown in students.

**Table 1 ijerph-19-09079-t001:** Sample description.

Variables	Women	Men	Total	
	Mean (Std)	Mean (Std)	Media (Std)	Student’s *t*-Test
Age	39.1 (17.0)	40.27 (18.9)	40.24 (17.0)	*p* < 0.001
Height (m)	1.64 (0.06)	1.76 (0.07)	1.67 (0.08)	*p* < 0.001
Weight (kg) in June	62.90 (12.00)	79.1 (12.85)	67.4 (14.2)	*p* < 0.001
BMI (Weight/Height) in June	23.19 (4.98)	25.4 (4.25)	23.8 (4.9)	*p* < 0.001
Hours of physical activity during lockdown/ week	4.3 (2.07)	4.75 (2.00)	4.43 (2.05)	*p* = 0.007
	**%**	**%**	**%**	**Chi-Square**
Physical activity during lockdown (%)				
Yes	69.8	69.6	69.8	*p* = 0.936
Not	30.2	30.4	30.2	
Educational level (%)				
High school or less	9.3	10.2	9.5	
College	21.8	25.4	22.8	*p* = 0.522
Graduate	68.9	64.4	67.7	
Occupation				
Qualified workers	42.7	32.4		*p* < 0.001
Low-skilled workers	19.1	22.3		
Student	23.5	18.2		
Housewives/Unemployed	3.4	2.2		
Retired	11.3	24.8		
Affected by COVID-19	13.9	11.6	13.2	*p* = 0.317
Hospitalized for COVID-19	1.1	0.3	0.9	*p* = 0.218

**Table 2 ijerph-19-09079-t002:** Frequency of consumption of vegetables, fruits, pulses, and dry fruits before and during lockdown by gender.

Before Lockdown					During Lockdown				Before/During (Kappa Test)		
Vegetables	Women *N* (%)	Men *N* (%)	Total *N* (%)	Chi-Square	Women *N* (%)	Men *N* (%)	Total *N* (%)	Chi- Square	Women	Men	Total
Never	9 (1.1)	7 (2.2)	16 (1.4)		14 (1.6)	9 (2.7)	23 (1.9)				
1-twice/week	387 (44.1)	161 (47.0)	548 (44.9)		356 (40.2)	150 (44.2)	506 (41.3)				
3–4 times/week	193 (22.0)	99 (28.8)	292 (23.9)	χ^2^ = 17.91	201 (22.7)	92 (27.1)	293 (23.9)	χ^2^ = 13.266	K = 0.613	K = 0.578	K = 0.607
5–6 times/week	96 (11.0)	30 (8.8)	126 (10.4)	*p* = 0.003	105 (11.9)	37 (10.9)	142 (11.6)	*p* = 0.021	*p* < 0.001	*p* < 0.001	*p* < 0.001
Once/day	98 (10.8)	25 (7.5)	123 (9.9)		105 (11.9)	24 (7.1)	129 (10.5)				
Twice or more/day	94 (11.0)	18 (5.6)	112 (9.6)		104 (11.8)	27 (8.0)	131 (10.7)				
**Fruits**											
Never	34 (4.0)	11 (3.5)	45 (3.9)		34 (3.9)	17 (5.0)	51 (4.2)				
1-twice/week	325 (38.4)	106 (33.9)	431 (37.2)		308 (35.0)	121 (35.8)	429 (35.3)				
3–4 times/week	110 (13.0)	49 (15.7)	159 (13.7)	χ^2^ = 5.544	124 (14.1)	42 (12.4)	166 (13.6)	χ^2^ = 2.108	K = 0.634	K = 0.631	K = 0.631
5–6 times/week	62 (7.3)	33 (10.5)	95 (8.2)	*p* = 0.353	78 (8.9)	30 (8.9)	108 (8.9)	*p* = 0.834	*p* < 0.001	*p* < 0.001	*p* < 0.001
Once/day	131 (15.5)	48 (15.3)	179 (15.4)		130 (14.8)	44 (13.0)	174 (14.3)				
Twice or more/day	185 (21.8)	66 (21.1)	251 (21.6)		205 (23.3)	84 (24.9)	289 (23.7)				
**Pulses**											
Never	40 (4.6)	14 (4.2)	54 (4.5)		42 (4.8)	19 (5.7)	61 (5.0)				
1-twice/week	679 (77.3)	244 (72.8)	923 (76.1)		589 (66.9)	204 (61.1)	793 (65.3)				
3–4 times/week	126 (14.4)	62 (18.5)	188 (15.5)	χ^2^ = 6.735	196 (22.2)	96 (28.7)	292 (24.0)	χ^2^ = 8.177	K = 0.490	K = 0.552	K = 0.509
5–6 times/week	24 (2.7)	12 (3.6)	36 (3.0)	*p* = 0.241	39 (4.4)	11 (3.3)	50 (4.1)	*p* = 0.147	*p* < 0.001	*p* < 0.001	*p* < 0.001
Once/day	4 (0.5)	3 (0.9)	7 (0.6)		11 (1.2)	4 (1.2)	15 (1.2)				
Twice or more/day	5 (0.6)	0 (0.0)	5 (0.4)		4 (0.5)	0 (0.0)	4 (0.3)				
**Nuts**											
Never	231 (26.4)	64 (19.6)	295 (24.5)		198 (22.7)	65 (19.5)	263 (21.8)				
1-twice/week	423 (48.3)	174 (53.2)	597 (49.7)		405 (46.4)	162 (48.5)	567 (47.0)				
3–4 times/week	106 (12.1)	48 (14.7)	154 (12.8)	χ^2^ = 7.343	130 (14.9)	47 (14.1)	177 (14.7)	χ^2^ = 3.307	K = 0.487	K = 0.530	K = 0.503
5–6 times/week	34 (3.9)	13 (4.0)	47 (3.9)	*p* = 0.196	48 (5.5)	25 (7.5)	73 (6.0)	*p* = 0.653	*p* < 0.001	*p* < 0.001	*p* < 0.001
Once/day	65 (7.4)	21 (6.4)	86 (7.2)		77 (8.8)	28 (8.4)	105 (8.7)				
Twice or more/day	16 (1.8)	7 (2.1)	23 (1.9)		15 (1.7)	7 (2.1)	22 (1.8)				

**Table 3 ijerph-19-09079-t003:** Frequency of consumption of foods of animal origin before and during lockdown by gender.

Before Lockdown					During Lockdown				Before/During (Kappa Test)		
Meat	Women *N* (%)	Men *N* (%)	Total *N* (%)	Chi-Square	Women *N* (%)	Men *N* (%)	Total *N* (%)	Chi-Square	Women	Men	Total
Never	47 (5.3)	12 (3.6)	59 (4.9)		44 (5.0)	15 (4.5)	59 (4.9)				
1-twice/week	355 (40.2)	134 (40.1)	489 (40.2)		356 (40.6)	122 (36.2)	478 (39.4)				
3–4 times/week	290 (32.9)	123 (36.8)	413 (34.0)	χ^2^ = 3.996	282 (32.2)	124 (36.8)	406 (33.4)	χ^2^ = 5.946	K = 0.675	K = 0.718	K = 0.686
5–6 times/week	125 (14.2)	39 (11.7)	164 (13.5)	*p* = 0.550	127 (14.5)	41 (12.2)	168 (13.8)	*p* = 0.312	*p* < 0.001	*p* < 0.001	*p* < 0.001
Once/day	55 (6.2)	21 (6.3)	76 (6.3)		56 (6.4)	28 (8.3)	84 (6.9)				
Twice or more/day	10 (1.1)	5 (1.5)	15 (1.2)		12 (1.4)	7 (2.1)	19 (1.6)				
**Fish**											
Never	67 (7.6)	25 (7.4)	92 (7.5)		79 (9.0)	26 (7.7)	105 (8.6)				
1-twice/week	558 (63.3)	214 (63.5)	772 (63.3)	χ^2^ = 1.241	513 (58.3)	203 (59.9)	716 (58.7)	χ^2^ = 5.237	K = 0.612	K = 0.602	K = 0.604
3–4 times/week	209 (23.7)	84 (24.9)	293 (24.0)	*p* = 0.941	223 (25.3)	90 (26.5)	313 (25.7)	*p* = 0.388	*p* < 0.001	*p* < 0.001	*p* < 0.001
5–6 times/week	36 (4.1)	11 (3.3)	47 (3.9)		52 (5.9)	17 (5.0)	69 (5.7)				
Once/day	11 (1.2)	3 (0.9)	14 (1.1)		13 (1.5)	2 (0.6)	15 (1.2)				
Twice or more/day	1 (0.1)	0 (0.0)	1 (0.1)		0 (0.0)	1 (0.3)	1 (0.1)				
**Eggs**											
Never	14 (1.6)	7 (2.1)	21 (1.7)		18 (2.0)	8 (2.4)	26 (2.1)				
1-twice/week	518 (59.2)	205 (61.6)	723 (59.9)	χ^2^ = 11.634	484 (54.9)	175 (51.6)	659 (54.0)	χ^2^ = 18.329	K = 0.671	K = 0.663	K = 0.664
3–4 times/week	269 (30.7)	84 (25.2)	353 (29.2)	*p* = 0.040	256 (29.0)	121 (35.7)	377 (30.9)	*p* = 0.003	*p* < 0.001	*p* < 0.001	*p* < 0.001
5–6 times/week	63 (7.2)	24 (7.2)	87 (7.2)		98 (11.1)	18 (5.3)	116 (9.5)				
Once/day	9 (1.0)	11 (3.3)	20 (1.7)		24 (2.7)	13 (3.8)	37 (3.0)				
Twice or more/day	2 (0.2)	2 (0.6)	4 (0.3)		2 (0.2)	4 (1.2)	6 (0.5)				

**Table 4 ijerph-19-09079-t004:** Frequency of consumption of dairy products before and during confinement by gender.

Before Lockdown					During Lockdown				Before/During (Kappa Test)		
Milk	Women *N* (%)	Men *N* (%)	Total *N* (%)	Chi-square	Women *N* (%)	Men *N* (%)	Total *N* (%)	Chi- Square	Women	Men	Total
Never	90 (10.4)	38 (11.3)	128 (10.6)		96 (11.0)	42 (12.4)	138 (11.4)				
1-twice/week	270 (31.1)	97 (29.0)	367 (30.5)		264 (30.2)	93 (27.4)	357 (29.4)				
3–4 times/week	57 (6.6)	24 (7.2)	81 (6.7)	χ^2^ =3.726	60 (6.9)	25 (7.4)	85 (7.0)	χ^2^ = 2.252	K = 0.753	K = 0.780	K = 0.759
5–6 times/week	48 (5.5)	27 (8.1)	75 (6.2)	*p* = 0.590	56 (6.4)	26 (7.7)	82 (6.8)	*p* = 0.813	*p* < 0.001	*p* < 0.001	*p* < 0.001
Once/day	268 (30.9)	103 (30.7)	371 (30.8)		261 (29.8)	105 (31.0)	366 (30.1)				
Twice or more/day	135 (15.6)	46 (13.7)	181 (15.0)		138 (15.8)	48 (14.2)	186 (15.3)				
**Yoghurt**											
Never	151 (17.4)	51 (15.7)	202 (17.0)		154(17.8)	56 (16.9)	210 (17.6)				
1-twice/week	356 (41.1)	124 (38.2)	480 (40.3)	χ^2^ = 9.266	323(37.4)	120 (36.3)	443 (37.1)	χ^2^ = 7.166	K = 0.702	K = 0.650	K = 0.689
3–4 times/week	88 (10.2)	49 (15.1)	137 (11.5)	*p* = 0.099	104(12.1)	47 (14.2)	151 (12.6)	*p* = 0.209	*p* < 0.001	*p* < 0.001	*p* < 0.001
5–6 times/week	58 (6.7)	29 (8.9)	87 (7.3)		63 (7.3)	29 (8.8)	92 (7.7)				
Once/day	161 (18.6)	59 (18.2)	220 (18.5)		157(18.2)	67 (20.2)	224 (18.8)				
Twice or more/day	52 (6.0)	13 (4.0)	65 (5.5)		62 (7.2)	12 (3.6)	74 (6.2)				
**Cheese**											
Never	98 (11.3)	25 (7.6)	123 (10.3)		101(11.7)	37 (11.1)	138 (11.5)				
1-twice/week	402 (46.3)	149 (45.6)	551 (46.1)	χ^2^ = 13.397	376(43.6)	139 (41.7)	515 (43.1)	χ^2^ = 10.205	K = 0.615	K = 0.683	K = 0.636
3–4 times/week	168 (19.3)	75 (22.9)	243 (20.3)	*p* = 0.020	159(18.4)	72 (21.6)	231 (19.3)	*p* = 0.070	*p* < 0.001	*p* < 0.001	*p* < 0.001
5–6 times/week	84 (9.7)	46 (14.1)	130 (10.9)		91 (10.6)	46 (13.8)	137 (11.5)				
Once/day	97 (11.2)	30 (9.2)	127 (10.6)		104(12.1)	36 (10.8)	140 (11.7)				
Twice or more/day	20 (2.3)	2 (0.6)	22 (1.8)		31 (3.6)	3 (0.9)	34 (2.8)				

**Table 5 ijerph-19-09079-t005:** Frequency of consumption of processed and industrial foods before and during lockdown by gender.

Before Lockdown					During Lockdown				Before/During (Kappa Test)		
Industrial Pastries	Women *N* (%)	Men *N* (%)	Total N(%)	Chi- square	Women *N* (%)	Men *N* (%)	Total *N* (%)	Chi- Square	Women	Men	Total
Never	433 (51.5)	153 (47.5)	586 (50.4)		455 (54.1)	174 (53.9)	629 (54.0)				
1-twice/week	311(37.0)	115 (35.7)	426 (36.7)		265 (31.5)	97 (30.0)	362 (31.1)				
3–4 times/week	57 (6.8)	33 (10.2)	90 (7.7)	χ^2^ = 9.391	67 (8.0)	34 (10.5)	101 (8.7)	χ^2^ = 2.360	K = 0.459	K = 0.487	K = 0.476
5–6 times/week	15 (1.8)	12 (3.7)	27 (2.3)	*p* = 0.094	24 (2.9)	7 (2.2)	31 (2.7)	*p* = 0.797	*p* < 0.001	*p* < 0.001	*p* < 0.001
Once/day	17 (2.0)	8 (2.5)	25 (2.2)		22 (2.6)	8 (2.5)	30 (2.6)				
Twice or more/day	7 (0.8)	1 (0.3)	8 (0.7)		8 (1.0)	3 (0.9)	11 (0.9)				
**Salty snacks**											
Never	275(32.2)	110 (34.3)	385 (32.8)		256 (29.9)	125 (38.3)	381 (32.2)				
1-twice/week	450(52.7)	161 (50.2)	611 (52.0)		398 (46.5)	131 (40.2)	529 (44.8)				
3–4 times/week	90 (10.5)	33 (10.3)	123 (10.5)	χ^2^ = 1.458	131 (15.3)	43 (13.2)	174 (14.7)	χ^2^ = 8.665	K = 0.389	K = 0.477	K = 0.415
5–6 times/week	26 (3.0)	10 (3.1)	36 (3.1)	*p* = 0.918	40 (4.7)	17 (5.2)	57 (4.8)	*p* = 0.123	*p* < 0.001	*p* < 0.001	*p* < 0.001
Once/day	12 (1.4)	6 (1.9)	18 (1.5)		27 (3.2)	8 (2.5)	35 (3.0)				
Twice or more/day	1 (0.1)	1 (0.3)	2 (0.2)		4 (0.5)	2 (0.6)	6 (0.5)				
**Fast-food**											
Never	305 (36.4)	105 (32.9)	410 (35.4)		436 (51.7)	164 (51.1)	600 (51.5)				
1-twice/week	471 (56.2)	187 (58.6)	658 (56.9)		339 (40.2)	129 (40.2)	468 (40.2)				
3–4 times/week	50 (6.0)	22 (6.9)	72 (6.2)	χ^2^ = 2.298	49 (5.8)	21 (6.5)	70 (6.0)	χ^2^ = 1.100	K = 0.387	K = 0.464	K = 0.408
5–6 times/week	7 (0.8)	4 (1.3)	11 (1.0)	*p* = 0.807	14 (1.7)	5 (1.6)	19 (1.6)	*p* = 0.954	*p* < 0.001	*p* < 0.001	*p* < 0.001
Once/day	4 (0.5)	1 (0.3)	5 (0.4)		4 (0.5)	2 (0.6)	6 (0.5)				
Twice or more/day	1 (0.1)	0 (0.0)	1 (0.1)		2 (0.2)	0 (0.0)	2 (0.2)				

**Table 6 ijerph-19-09079-t006:** Frequency of consumption of soft drinks and alcohol before and during lockdown by gender.

Before Lockdown					During Lockdown				Before/During (Kappa Test)		
Soft Drinks	Women *N* (%)	Men *N* (%)	Total *N* (%)	Chi- square	Women *N* (%)	Men *N* (%)	Total *N* (%)	Chi- Square	Women	Men	Total
Never	396 (46.9)	134 (42.0)	530 (45.6)	χ^2^ = 6.497 *p* = 0.261	440 (51.6)	150 (46.4)	590 (50.2)	χ^2^ = 4.942 *p* = 0.423	K = 0.512 *p* < 0.001	K = 0.589 *p* < 0.001	K = 0.534 *p* < 0.001
1-twice/week	335 (39.7)	126 (39.5)	461 (39.6)		265 (31.1)	105 (32.5)	370 (31.5)				
3–4 times/week	63 (7.5)	31 (9.7)	94 (8.1)		81 (9.5)	39 (12.1)	120 (10.2)				
5–6 times/week	20 (2.4)	11 (3.4)	31 (2.7)		32 (3.8)	12 (3.7)	44 (3.7)				
Once/day	19 (2.3)	13 (4.1)	32 (2.8)		19 (2.2)	12 (3.7)	31 (2.6)				
Twice or more/day	11 (1.3)	4 (1.3)	15 (1.3)		15 (1.8)	5 (1.5)	20 (1.7)				
**Beer/wine**				χ^2^ = 25.743 *p* < 0.001				χ^2^ = 15.023 *p* = 0.010	K = 0.478 *p* < 0.001	K = 0.517 *p* < 0.001	K = 0.501 *p* < 0.001
Never	317(37.6)	88 (27.6)	405 (34.9)		400 (46.7)	125 (38.5)	525 (44.4)				
1-twice/week	376(44.6)	138 (43.3)	514 (44.2)		295 (34.4)	108 (33.2)	403 (34.1)				
3–4 times/week	82 (9.7)	44 (13.8)	126 (10.8)		67 (7.8)	40 (12.3)	107 (9.1)				
5–6 times/week	30 (3.6)	17 (5.3)	47 (4.0)		28 (3.3)	16 (4.9)	44 (3.7)				
Once/day	30 (3.6)	21 (6.6)	51 (4.4)		50 (5.8)	23 (7.1)	73 (6.2)				
Twice or more/day	8 (0.9)	11 (3.4)	19 (1.6)		17 (2.0)	13 (4.0)	30 (2.5)				
**Distilled alcohol**				χ^2^ = 32.248 *p* < 0.001				χ^2^ = 19.599 *p* = 0.001	K = 0.354 *p* < 0.001	K = 0.462 *p* < 0.001	K = 0.404 *p* < 0.001
Never	644(78.2)	193 (62.9)	837 (74.0)		713 (86.0)	236 (75.2)	949 (83.0)				
1-twice/week	165 (20.0)	101 (32.9)	266 (23.5)		97 (11.7)	63 (20.1)	160 (14.0)				
3–4 times/week	12 (1.5)	9 (2.9)	21 (1.9)		12 (1.4)	10 (3.2)	22 (1.9)				
5–6 times/week	2 (0.2)	2 (0.7)	4 (0.4)		5 (0.6)	3 (1.0)	8 (0.7)				
Once/day	0 (0.0)	2 (0.7)	2 (0.2)		2 (0.2)	2 (0.6)	4 (0.3)				
Twice or more/day	1 (0.1)	0 (0.0)	1 (0.1)		0 (0.0)	0 (0.0)	0 (0.0)				

**Table 7 ijerph-19-09079-t007:** Inflammatory index of the diet before and during lockdown by gender.

Before Lockdown				During Lockdown			Before/During		
	Women	Men	T-Student	Women	Men	T-Student	T-Student for Related Samples		
	Mean (Std)	Mean (Std)		Mean (Std)	Mean (Std)		Women	Men	Total
**Legumes**	−1.33 (0.49)	−1.20 (0.45)	T = −4.421	−1.34 (0.51)	−1.24 (0.48)	T = −3.229	T = 1.145	T = 2.074	T = 2.044
			*p* < 0.001			*p* = 0.001	*p* = 0.253	*p* = 0.039	*p* = 0.041
**Fruits**	−0.67 (0.82)	−0.69 (0.81)	T = 0.345	−0.71 (0.83)	−0.73 (0.84)	T = 0.236	T = 2.445	T = 1.177	T = 2.682
			*p* = 0.730			*p* = 0.814	*p* = 0.015	*p* = 0.240	*p* = 0.007
**Pulses**	−1.00 (0.30)	−1.02 (0.32)	T = 0.929	−1.02 (0.34)	−1.01 (0.35)	T = −0.529	T = 1.670	T = −0.648	T = 1.053
			*p* = 0.353			*p* = 0.597	*p* = 0.095	*p* = 0.517	*p* = 0.293
**Nuts**	−0.37 (0.67)	−0.43 (0.70)	T = 1.309	−0.44 (0.70)	−0.46 (0.70)	T = 0.399	T = 4.067	T = 1.074	T = 3.993
			*p* = 0.191			*p* = 0.690	*p* < 0.001	*p* = 0.284	*p* < 0.001
**Meat**	0.63 (0.63)	0.67 (0.65)	T = −1.066	0.64 (0.64)	0.71 (0.66)	T = −1.842	T = −0.845	T = −1.961	T = −1.774
			*p* = 0.287			*p* = 0.066	*p* = 0.398	*p* = 0.051	*p* = 0.076
**Fish**	−0.98 (0.37)	−0.98 (0.36)	T = −0.204	−0.99 (0.41)	−0.99 (0.38)	T = −0.150	T = 0.703	T = 0.522	T = 0.870
			*p* = 0.838			*p* = 0.881	*p* = 0.482	*p* = 0.602	*p* = 0.384
**Eggs**	0.11 (0.36)	0.16 (0.44)	T = −1.811	0.15 (0.38)	0.13 (0.38)	T = 1.119	T = −3.250	T = 1.607	T = −1.790
			*p* = 0.071			*p* = 0.264	*p* = 0.001	*p* = 0.109	*p* = 0.074
**Milk**	0.65 (0.76)	0.61 (0.74)	T = 0.882	0.63 (0.76)	0.60 (0.73)	T = 0.656	T = 1.182	T = 0.239	T = 1.119
			*p* = 0.378			*p* = 0.512	*p* = 0.237	*p* = 0.811	*p* = 0.264
**Yoghurt**	0.35 (0.63)	0.34 (0.63)	T = 0.075	0.37 (0.65)	0.33 (0.59)	T = 1.169	T = −1.709	T = 0.750	T = −1.055
			*p* = 0.940			*p* = 0.243	*p* = 0.088	*p* = 0.454	*p* = 0.292
**Cheese**	0.20 (0.49)	0.18 (0.49)	T = 0.421	0.24 (0.56)	0.17 (0.45)	T = 2.376	T = −2.945	T = 0.478	T = −2.314
			*p* = 0.674			*p* = 0.018	*p* = 0.003	*p* = 0.633	*p* = 0.021
**Industrial pastries**	0.18 (0.55)	0.20 (0.57)	T = −0.630	0.20 (0.58)	0.19 (0.58)	T = 0.213	T = −1.290	T = 0.341	T = −0.933
			*p* = 0.529			*p* = 0.831	*p* = 0.197	*p* = 0.733	*p* = 0.351
**Salty snacks**	0.26 (0.60)	0.31 (0.66)	T = −1.221	0.38 (0.68)	0.37 (0.69)	T = 0.120	T = −5.495	T = −1.707	T = −5.549
			*p* = 0.222			*p* = 0.904	*p* < 0.001	*p* = 0.089	*p* < 0.001
**Soft drinks**	0.64 (0.63)	0.72 (0.66)	T = −2.107	0.58 (0.63)	0.66 (0.66)	T = −1.991	T = 3.140	T = 2.030	T = 3.729
			*p* = 0.035			*p* = 0.047	*p* = 0.002	*p* = 0.043	*p* < 0.001
**Fast food**	0.78 (0.65)	0.84 (0.65)	T = −1.315	0.63 (0.70)	0.66 (0.71)	T = −0.562	T = 6.674	T = 5.437	T = 8.488
			*p* = 0.189			*p* = 0.574	*p* < 0.001	*p* < 0.001	*p* < 0.001
**Total score**	−0.57 (2.85)	−0.28 (2.73)	T = −1.590	−0.68 (3.02)	−0.60 (2.89)	T = −0.417	T = 1.499	T = 2.548	T = 2.642
			*p* = 0.112			*p* = 0.677	*p* = 0.134	*p* = 0.011	*p* = 0.008

**Table 8 ijerph-19-09079-t008:** Differences in DII between occupations.

Differences between Occupation (H de Kruskal–Wallis)		
DII before lockdown	DII during lockdown	DII difference before-during
13.002	12.09	17.30
*p* = 0.011	*p* = 0.017	*p* = 0.002

## Data Availability

The datasets used and/or analysed during the current study available from the corresponding author on reasonable request.

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
