# Peer review of "Changes in Eating Behaviors during the COVID-19 Lockdown and the Impact on the Potential Inflammatory Effects of Diet"

_ijerph, 2022, doi:10.3390/ijerph19159079_

Round 1

Reviewer 1 Report

This manuscript describes a cross-sectional study of a Spanish non-probabilistic sample of adults comparing eating behaviors before and during the COVID-19 lockdown in March 2020 at the onset of the pandemic. The study found that in general, most people did not change their eating behavior and, contrary to the authors’ hypothesis, diet quality improved.  

The topic of the pandemic’s impact on a range of health-related behaviors, such as eating, substance use, and sleep, is incredibly important and it is heartening to see another paper adding to the growing literature aiming to capture the effect of a global public health crisis on an essential element of public health. However, there are several aspects of the study’s methodology and manuscript in need of revision and/or additional clarification.

1. One concern about this study is that all findings were analyzed and reported separately by sex. Presumably, this was important based on other pre-existing findings about sex differences in eating behavior (?), but there was no rationale for this decision provided in the introduction.

2. With regard to assessment, it would be useful for the authors to clarify the timeframe used in asking respondents about their eating habits before the start of the lockdown. Self-report dietary data has been criticized, with the main concern being that reporting errors may be so great as to render the data meaningless. Subar et al (2015, The Journal of Nutrition) assert that there is still value to using this type of data – and certainly the challenge of collecting data during a pandemic would be one instance when self-report data beats no data – but I think it’s important to acknowledge with more specificity the issues with self-report dietary data and underreporting patterns in the general population in the discussion section of the manuscript.   

3. Also with regard to assessment, can the authors clarify how they defined “physical activity at home” for participants?

4. In Table 2, what are “Pulses?” For all Tables, authors are encouraged to check the labels as some seem to be inaccurate and/or cut off.

5. This is an urban sample who, on average, had BMIs in the normal range. Can the authors clarify concerns about generalizing these findings to all of Spain based on the nation’s distribution of BMI and/or difference in lifestyle factors based on region?

In summary, the study addresses a very important topic and the authors are to be commended for quickly getting a survey out into their community during a time of intense stress and uncertainty. The manuscript, however, would be strengthened by clarification and revision as described above.  

Author Response

  1. One concern about this study is that all findings were analyzed and reported separately by sex. Presumably, this was important based on other pre-existing findings about sex differences in eating behavior (?), but there was no rationale for this decision provided in the introduction.

ANSWER:

First of all the decision to present the results disaggregated by gender has been following the recommendations of international organizations and scientific journals that insist on its importance.

Added one reference:

Tannenbaum C, Ellis RP, Eyssel F, Zou J, Schiebinger L. Sex and gender analysis improves science and engineering. Nature. 2019 Nov;575(7781):137–46.

On the other hand In the case of eating behavior in particular, there is an extensive literature on different patterns of eating behavior in men and women in different populations.

Added 2 references:

Shu-Hong Xu , Nan Qiao , Jian-Jun Huang , Chen-Ming Sun , Yan Cui , Shuang-Shuang Tian, Cong Wang , Xiao-Meng Liu , Hai-Xia Zhang , Hui Wang , Jie Liang , Qing Lu, Tong Wang Gender Differences in Dietary Patterns and Their Association with the Prevalence of Metabolic Syndrome among Chinese: A Cross-Sectional Study. 2016. Nutrients, 8, 180: 1-17.

Teresa Partearroyo , María de Lourdes Samaniego-Vaesken , Emma Ruiz , Javier Aranceta-Bartrina , Ángel Gil , Marcela González-Gross , Rosa M. Ortega , Lluis Serra-Majem, Gregorio Varela-Moreira. Current Food Consumption amongst the Spanish ANIBES Study Population. 2019. Nutrients 2019, 11, 2663;:1-15. https://doi.org/10.3390/nu11112663

LINE: 57-60

  1. With regard to assessment, it would be useful for the authors to clarify the timeframe used in asking respondents about their eating habits before the start of the lockdown. Self-report dietary data has been criticized, with the main concern being that reporting errors may be so great as to render the data meaningless. Subar et al (2015, The Journal of Nutrition) assert that there is still value to using this type of data – and certainly the challenge of collecting data during a pandemic would be one instance when self-report data beats no data – but I think it’s important to acknowledge with more specificity the issues with self-report dietary data and underreporting patterns in the general population in the discussion section of the manuscript.   

Answer:  Thank you very much for the suggestion, we recognize the importance of the comment and we added some paragraphs in the body of the text

LINE: 389-393

  1. Also with regard to assessment, can the authors clarify how they defined “physical activity at home” for participants?

Answer: A paragraph was added in Habits and lifestyle

Lines:159-162

  1. In Table 2, what are “Pulses?” For all Tables, authors are encouraged to check the labels as some seem to be inaccurate and/or cut off.

Answer: Pulses (dry beans, dry peas, lentils and chickpeas) are nutrient dense foods that possess many beneficial effects. They are also among the most versatile and culturally diverse foods in the world, acting as a staple protein in certain countries. Although global pulse production has remained steady at around 40 million tonnes per year, in the Western world, consumption rates remain low in comparison to Asian and African markets, with Canada leading the global export of pulse crops. Future work directed at the long term benefits of pulse consumption on diet quality and disease reduction coupled with efforts targeted toward increasing global consumption rates are critical to reinforce the role of pulses in a healthy diet

Leterme P, Carmenza Muñoz L. Factors influencing pulse consumption in Latin America. Br J Nutr. 2002 Dec;88 Suppl 3:S251-5. doi: 10.1079/BJN/2002714. PMID: 12498624.

  1. This is an urban sample who, on average, had BMIs in the normal range. Can the authors clarify concerns about generalizing these findings to all of Spain based on the nation’s distribution of BMI and/or difference in lifestyle factors based on region?

Answer: Added a paragraph and a reference to the thread about your suggestion and a reference

Aranceta Bartrina J, Pérez Rodrigo C. Desigualdad, salud y nutrición en España: una visión regional del índice de masa corporal. Nutr Hosp 2018;35(N.º Extra. 5):142-149

Lines: 327-331

In summary, the study addresses a very important topic and the authors are to be commended for quickly getting a survey out into their community during a time of intense stress and uncertainty. The manuscript, however, would be strengthened by clarification and revision as described above.  

Reviewer 2 Report

The manuscript entitled “Changes in eating behaviors during the COVID-19 lockdown and the impact on the potential inflammatory effects of diet” by Montero Lopez and colleagues provides a good piece of work that highlights one of the important aspects of daily habits and behaviors that significantly changed as a consequence of the pandemic. Overall, this paper shows interesting data that fits into the topic of the journal.

The authors presented the data from the ALVIMED project on the impact of confinement on diet quality in men and women.

The manuscript is clearly written and the answers to the questions and discussion were short and precise., however, I have major concerns regarding the data analysis and methodology that should be given serious consideration by the authors.

The authors focused on gender-associated eating behavior, the data are presented and results are comparing males vs females while withing each gender the groups were stratified and included subjects from with different incomes, economic status as well as educational levels. The authors need to analyze their data taking these sub-groups/stratifications into consideration.

The presentation/analysis of data, discussion, and conclusions must take into consideration these sub-groups and stratification for a better understanding of the effect of COVID on eating behaviors in different socioeconomic groups, and work statuses… not only males vs females.  A simple T-test wouldn’t be enough to show the statistical difference between the groups in this big data.

Author Response

Referee 2

Comments and Suggestions for Authors

The manuscript entitled “Changes in eating behaviors during the COVID-19 lockdown and the impact on the potential inflammatory effects of diet” by Montero Lopez and colleagues provides a good piece of work that highlights one of the important aspects of daily habits and behaviors that significantly changed as a consequence of the pandemic.

 Overall, this paper shows interesting data that fits into the topic of the journal. The authors presented the data from the ALVIMED project on the impact of confinement on diet quality in men and women. The manuscript is clearly written and the answers to the questions and discussion were short and precise., however, I have major concerns regarding the data analysis and methodology that should be given serious consideration by the authors. The authors focused on gender-associated eating behavior, the data are presented and results are comparing males vs females while withing each gender the groups were stratified and included subjects from with different incomes, economic status as well as educational levels.

The authors need to analyze their data taking these sub-groups/stratifications into consideration. The presentation/analysis of data, discussion, and conclusions must take into consideration these sub-groups and stratification for a better understanding of the effect of COVID on eating behaviors in different socioeconomic groups, and work statuses… not only males vs females. A simple T-test wouldn’t be enough to show the statistical difference between the groups in this big data.

Answer

Thank you very much for your comments, in our study the index of inflammation of the diet is a score, a continuous variable. The objective of the study, given its design, was not to associate this index with lifestyle or health indicators. Our objective was to verify if the change in eating behavior before and during confinement had an impact on the anti-inflammatory or pro-inflammatory potential of the diet.

The decision to present the results disaggregated by gender has been following the recommendations of international organizations and scientific journals that insist on its importance.

Reference: Tannenbaum C, Ellis RP, Eyssel F, Zou J, Schiebinger L. Sex and gender analysis improves science and engineering. Nature. 2019 Nov;575(7781):137–46.

On the other hand, in the case of eating behavior, there is extensive literature on different patterns of eating behavior in men and women in different populations, for this reason we found it very convenient to disaggregate the results by gender

Added 2 references:

Teresa Partearroyo , María de Lourdes Samaniego-Vaesken , Emma Ruiz , Javier Aranceta-Bartrina , Ángel Gil , Marcela González-Gross , Rosa M. Ortega , Lluis Serra-Majem, Gregorio Varela-Moreira. Current Food Consumption amongst the Spanish ANIBES Study Population. 2019. Nutrients 2019, 11, 2663;:1-15. https://doi.org/10.3390/nu11112663.

Shu-Hong Xu , Nan Qiao , Jian-Jun Huang , Chen-Ming Sun , Yan Cui , Shuang-Shuang Tian, Cong Wang , Xiao-Meng Liu , Hai-Xia Zhang , Hui Wang , Jie Liang , Qing Lu, Tong Wang Gender Differences in Dietary Patterns and Their Association with the Prevalence of Metabolic Syndrome among Chinese: A Cross-Sectional Study. 2016. Nutrients, 8, 180: 1-17.

LINES: 57-60

Reviewer 3 Report

Manuscript ID: ijerph-1774990

Changes in eating behaviors during the COVID-19 lockdown and the impact on the potential inflammatory effects of diet

Authors: María del Pilar Montero López, Ana Isabel Mora Urda, Francisco Javier Martín Almena and Oscar Geovanny Enríquez-Martínez

General comment to the authors

The manuscript's authors studied the dietary habits before and during Covid19 lockdown in Spain by means of a survey. They found that during lockdown, the diet of the respondents was healthier (less pro-inflammatory), mainly due to an increase in the consumption of legumes and fruits together with a decrease in processed food, fast food, soft drinks and alcoholic beverages. The methodology is correct, the authors themselves mention the weaknesses, and the conclusions are coherent. However, some aspects need to be improved. Furthermore, it is worth mentioning that the authors mention supplementary material to which the reviewer has not had access.

Specific Comments

- Introduction:

First paragraph need reference or references. The same for the second paragraph in relation with the sentence that WHO decreed Covid-19 as a pandemic, the WHO document must be cited.

Paragraph six: After “…(TNFα)” also need reference(s). Next sentence “CRP is…..cell adhesion” is not understood and should be rewritten. The authors refer to what is described in reference 3 pg 596 section 4, but it is not really understood as it is expressed here.

In the reviewer’s opinion, the last paragraphs since “this work” to the end of introduction should be written as a single paragraph.

- Materials and methods:

   Section ”dietary inflammation index (DII)” first paragraph the reference 10 seems to be incorrect, it could be the 8?

BMI is weight (kg) / height2 (m2), not “weight (kg) / height2 (m)”.

In the antepenultimate paragraph, the sentence “According to…..than 0.81”, again de reference 10 seems incorrect.

- Results:

The headings of the "Eating behavior" and "Dietary inflammation Index (DII)" sections are not in the same format and this confuses the reader.

   Tables 5 and 6 do not match the headings with the content. They are interchanged.

The sentence referring to Table 7 in the first paragraph concerning DII "...there were differences between women and men,..." is not quite accurate, the total score is not statistically different before and after between men and women, only if comparing before/after. It should be rewritten.

Further on, the sentence "During lockdown, the DII became more negative for both genders, which reflects that the consumption of foods with anti-inflammatory potential increased, and the difference be-tween the genders decreased." would be true if it is commented that there are no significant differences in any case. This appears to be observed, but is not significant!!!!. As commented below, there are only differences between before and during lockdown for men, not for women. This should be described more clearly by highlighting what is really significant.

In the materials and methods, the alcohol is included for the DII calculation, so why do the authors not include alcohol in the table7?

Figure 1: in the text the authors mentioned figure 1, but the figure has no figure caption indicating the number neither what show the figure!!!! Furthermore, in the figure must be the symbols indicating the significant differences between groups that the authors mentioned in the text.

The affirmation Statistically significant differences were observed between the four occupational groups considered, with housewives, unemployed people and retired showing the most anti-inflammatory food values both before and during confinement (Table 8).” I am unable to see this in this table!!! Globally yes, but among them? Please, arrange figure 1 and table 8 for a correct understanding.

- Discussion:

Where does the decision start? There is no title!!!

The sentence Therefore, our initial hypothesis is not fulfilled” alone, have not any sense

   In general, there are too many small paragraphs that should be together, e.g., "In this sample..." + "In general..." + "therefore..." would be one paragraph.

Finally, authors should make a general revision of the text of all manuscript of both in scientific style and in English.

Author Response

General comment to the authors

The manuscript's authors studied the dietary habits before and during Covid19 lockdown in Spain by means of a survey. They found that during lockdown, the diet of the respondents was healthier (less pro-inflammatory), mainly due to an increase in the consumption of legumes and fruits together with a decrease in processed food, fast food, soft drinks and alcoholic beverages. The methodology is correct, the authors themselves mention the weaknesses, and the conclusions are coherent. However, some aspects need to be improved. Furthermore, it is worth mentioning that the authors mention supplementary material to which the reviewer has not had access.

Specific Comments

- Introduction:

  1. First paragraph need reference or references. The same for the second paragraph in relation with the sentence that WHO decreed Covid-19 as a pandemic, the WHO document must be cited.

Answer: Done. Two references have been added, in the text and in the bibliography section, reference 1 and 2

Lines 35-38

  1. Paragraph six: After “…(TNFα)” also need reference(s). Next sentence “CRP is…..cell adhesion” is not understood and should be rewritten. The authors refer to what is described in reference 3 pg 596 section 4, but it is not really understood as it is expressed here.

Answer:  Added reference after ‘ …TNFα’¡(Da Silva et al., 2015).  The sentence has been rewritten.

Bibliographic citations have been renumbered.

Lines 64-69

  1. In the reviewer’s opinion, the last paragraphs since “this work” to the end of introduction should be written as a single paragraph.

Answer:

Done in part. We consider that the objectives should go outside the paragraph to make them clearer

Lines 86-95

- Materials and methods:

  1.    Section ”dietary inflammation index (DII)” first paragraph the reference 10 seems to be incorrect, it could be the 8?

Answer: It is indeed a mistake. Bibliographic citations have been renumbered.

  1. BMI is weight (kg) / height2 (m2), not “weight (kg) / height2 (m)”.

Answer: It is indeed a mistake, it has been corrected in the text

Lines 163-164

  1. In the antepenultimate paragraph, the sentence “According to…..than 0.81”, again de reference 10 seems incorrect.

Answer: It is indeed a mistake. Bibliographic citations have been renumbered.

- Results:

  1. The headings of the "Eating behavior" and "Dietary inflammation Index (DII)" sections are not in the same format and this confuses the reader.

Answer: Correct, the two headings have been put with the same format

Line 202

  1.  Tables 5 and 6 do not match the headings with the content. They are interchanged.

Answer: The titles were exchanged when copying the tables in the Template. It has been corrected

Lines 559, 591

  1. The sentence referring to Table 7 in the first paragraph concerning DII "...there were differences between women and men,..." is not quite accurate, the total score is not statistically different before and after between men and women, only if comparing before/after. It should be rewritten.
  1. Answer: The sentence and the following paragraphs have been rewritten

Lines 282-283

  1. Further on, the sentence "During lockdown, the DII became more negative for both genders, which reflects that the consumption of foods with anti-inflammatory potential increased, and the difference be-tween the genders decreased." would be true if it is commented that there are no significant differences in any case. This appears to be observed, but is not significant!!!!. As commented below, there are only differences between before and during lockdown for men, not for women. This should be described more clearly by highlighting what is really significant.
  1. Answer: The paragraphs have been rewritten

Lines 290 -296

  1. In the materials and methods, the alcohol is included for the DII calculation, so why do the authors not include alcohol in the table7?

Answer:  The calculation of the inflammation score was performed following the instructions of Davis et al. 2019. They categorised each item as either anti-inflammatory (e.g. fish consumption) or pro-inflammatory (e.g. red meat consumption) from their reported associations with CRP. Then they assigned each item's response options a value from −2 (anti-inflammatory) to +2 (pro-inflammatory). Finally, they summed all items to generate an inflammatory diet score for each participant where higher scores indicate a more pro inflammatory diet.

The authors did not consider alcohol to calculate the DII score.

In our study information on alcohol consumption was collected but it was not used in the calculation of the inflammation index. It is a mistake and alcohol has been removed in the calculation section of the DII of Material and Methods.

Line 148

  1.  Figure 1: in the text the authors mentioned figure 1, but the figure has no figure caption indicating the number neither what show the figure!!!! Furthermore, in the figure must be the symbols indicating the significant differences between groups that the authors mentioned in the text.

Answer: Our apologies, the figure title was not pasted into the template. We have also changed the type of graph to make the results clearer.

Line 662

  1. The affirmation “Statistically significant differences were observed between the four occupational groups considered, with housewives, unemployed people and retired showing the most anti-inflammatory food values both before and during confinement (Table 8).” I am unable to see this in this table!!! Globally yes, but among them? Please, arrange figure 1 and table 8 for a correct understanding.

Answer: We have changed the type of graph to make the results clearer and added

- Discussion:

  1.  Where does the decision start? There is no title!!!

Answer: The discussion section has been numbered and put in bold

Line 322

  1. The sentence “Therefore, our initial hypothesis is not fulfilled” alone, have not any sense

Answer: Paragraph has been rewritten

Lines 334-341

  1. In general, there are too many small paragraphs that should be together, e.g., "In this sample..." + "In general..." + "therefore..." would be one paragraph.

Answer: We have put together some paragraphs

Finally, authors should make a general revision of the text of all manuscript of both in scientific style and in English.

Answer: The style of the manuscript was revised by American Journal Expert. We attach the editing certificate.

Reviewer 4 Report

This cross-sectional study aims at describing food behaviors and meal characteristics during COVID-19 pandemic. Although of potential interest, the manuscript raises several limitations that need to be adequately addressed to this reviewer.

1. Cavicchia index is vaguely referenced ([9] or [10])?

2. One sole question on physical activity levels is not diriment and must be assessed through a validated questionnaire (which I doubted has been used).

3. There are various controlling factors that need to be taken into account in order to make whatever inferences.

a) Inflammation was not blood-derived.

b) FFQI is a self reported measure

c) google forms interviews are subjected to recall bias 

4. The statistical approach (and, especially its power) is frankly weak and one prove of this comes from the difficulty to test your working hypothesis.

5. The overall design relies on one association and therefore it runs the risk to be over-simplistic. 

6. A proinflammatory diet is one that favours increased levels of proinflammatory biomarkers, such as C-Reactive protein (CRP), interleukins 6 and 8 (IL-6 and IL-8, respectively) and tumour necrosis factor alpha (TNF-αa)This passage needs to be referenced with studies indicating increase in these factors upon “inflammatory” diets. Reference suggestion: https://doi.org/10.1093/cdn/nzy098.

7. The paper seems to lack of a proper discussion (which seems to be partly embedded in the “results” section). For clarity and in the interests of reader, the authors are suggested to report a “standard” discussion after the section “results”.

8. Was the DII compared with the results obtained against any reference method (for example Cavicchia et al)?

Author Response

This cross-sectional study aims at describing food behaviors and meal characteristics during COVID-19 pandemic. Although of potential interest, the manuscript raises several limitations that need to be adequately addressed to this reviewer.

  1. Cavicchia index is vaguely referenced ([9] or [10])?

Answer: Bibliographic citations have been renumbered.

Line 445-511

  1. One sole question on physical activity levels is not diriment and must be assessed through a validated questionnaire (which I doubted has been used).

Answer: It was not an objective of the study to analyze the formal physical activity carried out by each person in terms of exercise, regular sport, etc. Therefore, no validated questionnaire was used to know the level of physical activity.

  1. There are various controlling factors that need to be taken into account in order to make whatever inferences.
  2. a) Inflammation was not blood-derived.

Answer: No blood samples were collected. People were confined to their homes.

For this reason, we used the Index developed by Cavicchia et al., with the aim of validating the relationship between the inflammatory index with biochemical blood samples and food consumption.

On the other hand, the pandemic situation did not allow the collection of biological samples.

  1. b) FFQI is a self reported measure

Answer: It was not possible to ask personally about consumption since people were confined. For this reason, a validated questionnaire was used that was turned into a google forms, as in many other investigations carried out during confinement due to the pandemic, In addition, it is cited as a limitation.

  1. c) google forms interviews are subjected to recall bias 

Answer: The authors are aware of the limitations of online questionnaires. As in other works cited in the bibliography, this method was used due to people's mobility limitations, specifically in Spain there was a total lockdown for 4 months between March and June 2020. Limitations of the study are discussed in the discussion.

  1. The statistical approach (and, especially its power) is frankly weak and one prove of this comes from the difficulty to test your working hypothesis.

Answer: We do not agree with this statement.

The normality of the variables was verified, in the case of non-compliance, non-parametric tests were used.

The kappa test was also used to measure the degree of concordance between the responses of the participants on the food consumption they had before confinement and the one they were having at the time of confinement.

     Our hypothesis was based on data reported by the spanish ministry of consumptionpurchases of food and other products at the start of the pandemic.  The general idea was that the diet was going to get worse, but later it has been seen in this and other studies that this was not the case. In samples of medium-high sociocultural level, eating behavior improved. Added a paragraph to the discussion and a bibliographic citation

Lines 334-341

Added one reference: Ministerio de agricultura  pesca y alimentación. Informe del consumo de alimentación en España 2020. 2021. 1–740 p.

  1. The overall design relies on one association and therefore it runs the risk to be over-simplistic. 

Answer: We appreciate the comment. Indeed, it is a cross-sectional study that was carried out in very difficult conditions of isolation from the population due to confinement. Even so, we think that both the design of the study and the variables considered allow us to assess the change in eating behavior due to confinement, differences between men and women and differences between occupations, and analyze its relationship with an already established index. We consider that these contributions are important to be published in the special issue of IJERPH dedicated to eating behavior..

  1. “A proinflammatory diet is one that favours increased levels of proinflammatory biomarkers, such as C-Reactive protein (CRP), interleukins 6 and 8 (IL-6 and IL-8, respectively) and tumour necrosis factor alpha (TNF-αa)”. This passage needs to be referenced with studies indicating increase in these factors upon “inflammatory” diets. Reference suggestion: https://doi.org/10.1093/cdn/nzy098.

Answer: We thank the reviewer for this suggestion of this most recent bibliographical citation. We have added it to the text and in the references section.

  1. The paper seems to lack of a proper discussion (which seems to be partly embedded in the “results” section). For clarity and in the interests of reader, the authors are suggested to report a “standard” discussion after the section “results”.

Answer: Following your indications we have incorporated into the discussion some paragraphs of the results.

Lines 323-331

  1. Was the DII compared with the results obtained against any reference method (for example Cavicchia et al)?

Answer: Information on frequency of food consumption was used following the indications of Davis et al., 2019

Round 2

Reviewer 2 Report

No additional comments.

Reviewer 4 Report

I thank the Authors for having sufficiently addressed my comments. Some issues remained unsolved, however the limitation paragraph provides the magnitude of the limits and the interpretative difficulties of this study.

This manuscript is a resubmission of an earlier submission. The following is a list of the peer review reports and author responses from that submission.

Round 1

Reviewer 1 Report

The manuscript entitled „Changes in nutritional behaviours during the COVID-19 lock-down and the impact on the potential inflammatory effects of diet” presents interesting data, however in my opinion manuscript  it contains serious flaws. Moreover, the quality of the presented manuscript requires major corrections.

Introduction

  • Please complete the references in the first two paragraphs.
  • Line 65-67 What other products contain saturated fatty acids and simple sugars?
  • Line 78-93 The text needs to be improved.
  • The Introduction section needs better justification.
  • What is the current state of knowledge? Are there any research findings on a pro-inflammatory diet during the COVID-19 pandemic?
  • Is the following study a continuation of similar studies conducted during the pandemic?

Materials and methods

  • This section requires significant editorial and content-related corrections.
  • Please describe in detail the recruitment of participants - what were the inclusion and exclusion criteria?
  • What questionnaires were used in this study? Did Authors validate the questionnaires before?
  • Students of which fields of study took part in the study? What was the% of students in the study, faculty members, administrative staff and other participants unrelated to the university?
  • How was the sample size for the test calculated? Please complete this.
  • Did Authors conducted a pilot study?
  • Please provide the age range for the respondents. It is only known that they were people over 18 years of age.
  • There are no details about the study. When was the study conducted? Were data collected twice before and during the pandemic?
  • What questionnaire was used to assess the level of physical activity?
  • The question of being active at home is not enough to assess physical activity.
  • Was it possible to take up outdoor activities during the pandemic?
  • Was it forbidden to leave the house during the entire pandemic period?
  • What was the status health of the study participants? What diseases occurred in the participants?
  • What was the diet of the participants at the start of the study?
  • Were the participants on any diets, e.g. vegetarian? This could affect the results obtained.
  • How many questionnaires were excluded from the analysis? How many participants were excluded and for what reasons? Please provide a study scheme.

Results

  • This section requires major corrections.
  • Table 1 – Does it concern the characteristics of the group before or during the pandemic?
  • Table 1 – error in the age of men (4 years)?
  • Table 1 – the point about physical activity is incomprehensible.
  • Table 1 – the educational point is incomprehensible - please provide all the answers.
  • Line 188-193 - In the description of the results, please indicate which results were statistically significant.
  • Please add to the description of results the differences between the group stratified by changes in body weight, age and place of residence, level of physical activity. What was the impact of these factors on the changes in the diet of the study participants? Giving the differences only between men and women is not enough. Please provide the tables in the Supplementary File.
  • Women N (%) – what does N mean?
  • There are no descriptions under the tables - please complete it.
  • How did Authors assess that men increased their vegetable consumption during the pandemic? The percentage of responses is similar. Perhaps the structure of the table makes it difficult to analyze the results. It would be advisable to rebuild the layout of the tables.
  • Please modify the description of the results and describe the specific statistically significant differences.

Discussion

Discussion is the weakest element of the manuscript. Conducted in a chaotic manner, without taking into account the most important statistically significant results, and above all without discussing the results concerning the pro-inflammatory diet.

Line 364-365 – What were the results in other studies?

Line 366-371 – Authors compared women with men, there are no comparisons depending on the level of education, so this paragraph is inadequate to the analyzed data.

Line 373-377 – Did Authors ask the respondents about how they spend their time, how they prepare meals? Did Authors take into account the inhomogeneity of the study group (students, lecturers, people not related to the university)?

Line 378-381 – Please complete the results with the division of groups depending on age, body weight and physical activity.

Line 382-385 – No comparison with the results of other authors, no explanation of the results obtained.

Line 394-399 – This is a serious limitation of the presented study and requires detailed discussion. Are other studies showing a decrease in activity levels during the pandemic? How can changes in physical activity also affect eating behavior? Please complete this.

Line 403 - How did Authors assess the level of participants' activity before and during the pandemic?

Line 409-415 – this fragment of the text does not refer to the parameters analyzed in the study.

Conclusions

Conclusions need to be corrected after re-analysis the data.

References

Requires editorial improvement.

Reviewer 2 Report

As the authors note, investigations into how diet quality changed during the COVID-19 lockdowns are needed, and this study does present some data related to this question. However, there are several key concerns about the study that make it difficult to know how reliable the results are. Two major concerns are:

1) The sample appears to be purely a convenience sample and may be very skewed. It's not quite clear who is really represented in the sample - how were the initial students/teachers/staff at the university approached? Even if there was a general recruitment approach throughout the university (and not, say, just a select group that the authors already knew, or members of a particular labor department) It seems likely that these students/teacher/staff and their friends would be individuals who would be more likely to be interested in a study on diet quality / more highly motivated to have higher quality diets in the first place.  So it is not clear that the results can be generalized at all to any other sample. 

2) while the authors constructed a scale to measure the inflammatory potential of the diet that has a basis in the literature, it does not appear that any validated measures were used for any of the variables - e.g. was a validated food frequency questionnaire used to measure dietary frequencies? How was physical activity assessed? Given that measuring diet and physical activity accurately is extraordinarily difficult, and that using non-validated measures can often result in large amounts of measurement error (whether through random error and/or systematic under- and over-reporting due to social desirability biases), the authors need to persuade the reader that sound measurement approaches were used. This is also a very large issue for the measures of weight and height, as well as weight loss, all of which are subject to substantial bias when measured via self-report. 

There are some other concerns in regards to the strength of the language used. For example, the authors should not use terminology in this paper that implies strong causal relationships were determined, such as “effect” or “impact.” I’d suggest using terms like “association” or “associated with” only. Even with a stronger sampling design and validated measures, the study is observational, so it would be impossible to determine causality. Relatedly, because diet and weight were not measured before and after the pandemic, there is likely substantial reporting bias on the part of the participants, so in the best case scenario the authors really can only report on people’s perceived changes during COVID.

I had some other comments and suggestions in the attached, but I do think the issue of the skewed sample and the lack of sound measures are overwhelmingly the issues to focus on.

Reviewer 3 Report

All variables of this study are well processed and the results seem reliable. 

Nevertheless, a major weakness is that no blood samples to measure CRP, IL-6 and other parameters have been taken. Hence, the data remain diffuse concerning hard facts. On the other hand, the participants are of higher educational level. So their responses may be more believable in a certain extent.

This referee recommends to mention this fact in a section "limitations".

Minor:

Table 1, age of men. The given number is unrealitic (4.27)

Round 2

Reviewer 1 Report

I uphold the decision to reject this manuscript, because Authors conducted a study among people related to the topic of nutrition, and this undoubtedly has an impact on the obtained results, which do not reflect the situation in the general population.  If Authors provided additional results for the control group, the manuscript could be published. However, in its current version, this study does not bring any new knowledge about the impact of the pandemic on the diet of the population. Additionally, the selection of the study group is incorrect and may lead to incorrect results. 

Reviewer 2 Report

This study still has some concerning flaws that are not appropriately addressed in the limitations section or elsewhere (e.g. the concept of "change" is still referred to even though, at best, a cross-sectional study can only ever measure perceived change). Several comments I had raised were not addressed.